# Identification of peptides from honeybee gut symbionts as potential antimicrobial agents against *Melissococcus plutonius*

Haoyu Lang [1], Yuwen Liu[1], Huijuan Duan[1], Wenhao Zhang[1], Xiaosong Hu[1] & Hao Zheng [1] ✉

Eusocial pollinators are crucial elements in global agriculture. The honeybees and bumblebees are associated with a simple yet host-restricted gut community, which protect the hosts against pathogen infections. Recent genome mining has led to the discovery of biosynthesis pathways of bioactive natural products mediating microbe-microbe interactions from the gut microbiota. Here, we investigate the diversity of biosynthetic gene clusters in the bee gut microbiota by analyzing 477 genomes from cultivated bacteria and metagenome-assembled genomes. We identify 744 biosynthetic gene clusters (BGCs) covering multiple chemical classes. While gene clusters for the post-translationally modified peptides are widely distributed in the bee guts, the distribution of the BGC classes varies significantly in different bee species among geographic locations, which is attributed to the strain-level variation of bee gut members in the chemical repertoire. Interestingly, we find that *Gilliamella* strains possessing a thiopeptide-like BGC show potent activity against the pathogenic *Melissococcus plutonius*. The spectrometry-guided genome mining reveals a RiPP-encoding BGC from *Gilliamella* with a 10 amino acid-long core peptide exhibiting antibacterial potentials. This study illustrates the widespread small-molecule-encoding BGCs in the bee gut symbionts and provides insights into the bacteria-derived natural products as potential antimicrobial agents against pathogenic infections.

The honeybees (*Apis mellifera*), which pollinate various plant species, significantly impact biodiversity and are thus considered a crucial component of ecosystems. In addition to their vital role in pollination, honeybees possess significant agronomic and economic value due to their ability to produce various valuable commercial products[1]. However, managed honeybee colonies have decreased globally in recent decades, known as colony collapse disorder (CCD)[2]. The colony loss has been described as multifactorial, involving combinations of environmental stress and infectious agents[3,4]. Honeybees are susceptible to various biological agents that cause disease, including eukaryotic[5,6] and prokaryotic pathogens[7]. Bacterial diseases are a significant contributing factor to the colony collapse disorder of

honeybee populations[2]. American Foulbrood[8] and European Foulbrood (EFB)[9] are the two primary bacterial diseases, mainly attacking bees during the larval stage[10]. This can lead to immense brood loss and may result in weakened colonies and colony collapse[11]. EFB is a brood disease caused by the bacterial pathogen *Melissococcus plutonius* that multiplies in the midgut, resulting in nutrient deprivation and, ultimately, the death of infected larvae[11].

Although adult bees do not suffer from EFB, they can be asymptomatic carriers transmitting *M. plutonius* through contaminated food to bee larvae within colonies[12]. To prevent the spread of *M. plutonius*, antibiotics are prophylactically adopted for treating EFB[9]. In North America, antimicrobials are used to control brood diseases, and

[1]College of Food Science and Nutritional Engineering, China Agricultural University, 100083 Beijing, China. ✉e-mail: hao.zheng@cau.edu.cn

oxytetracycline is the approved antibiotic for treating EFB diseases[13]. However, with the growing concern of antibiotic contamination of bee products, prophylactic antimicrobial use is prohibited in beekeeping in the European Union[13]. In addition, antibiotic resistance has been observed in the *M. plutonius*, which makes infections harder to treat and increases the risk of disease spread[13]. Moreover, antibiotic treatments disrupted the native gut microbiota, leading to a weakened immune system that makes honeybees more prone to opportunistic pathogens[14]. The use of antibiotics also contributes to the dissemination of antibiotic-resistance genes among native gut bacteria[15].

Both honey (genus *Apis*) and bumble bees (genus *Bombus*) harbor simple yet highly specialized microbial communities in their guts. The bee gut microbiota comprises a limited number of core members, which include two genera of lactic acid bacteria, *Lactobacillus* Firm5 and *Bombilactobacillus*, *Gilliamella*, *Snodgrassella*, and *Bifidobacterium*. Notably, most bee gut bacteria contain several divergent lineages and a great extent of strain-level diversity[16,17]. It has been shown that bee gut microbiota is critical in host nutrition, weight gain, endocrine signaling, and bee social behaviors[18]. Bee gut bacteria also possess immunomodulation effects[16] and protect hosts from opportunistic pathogens[19]. *Lactobacillus apis* stimulate the host Toll signaling pathway and the downstream expression of AMP genes, and the produced apidaecin exhibited a high inhibitory effect on the pathogen, *Hafnia alvei*[19]. The bee gut bacteria can enhance the host's immune response, but it has been unclear whether they produce bioactive molecules mediating microbe-microbe interactions.

It has been found that the symbiotic microbiota of variable hosts, including insects[20], mammals[21], and plants[22], has a strong potential to synthesize large amounts of bioactive compounds. These structurally distinct secondary metabolites always exert antibacterial activity, which protects the host from external pathogens[23]. For example, the Ruminococcin A produced by *Ruminococcus gnavus* isolated from the human gut has exhibited antibacterial activity[24]. Zwittermicin A is an antibiotic identified from *Bacillus cereus* with broad-spectrum activity against Gram-positive and Gram-negative prokaryotic microorganisms[25]. Discovering secondary metabolites from the gut microbiota may provide a promising strategy for improving resistance against bacterial diseases.

The secondary metabolites are often small chemical compounds produced by biosynthetic enzymes encoded by biosynthetic gene clusters (BGCs). Recent studies identified potential BGCs in honeybee gut *Apilactobacillus kunkeei*[26] and *Frischella perrara*[27]. These secondary metabolites are thought to mediate microbe-microbe and microbe-host interactions. *F. perrara* harbors a BGC for the biosynthesis of aryl polyene, which may assist its persistence in the pylorus by resisting reactive oxygen species from the host immune system[27]. It is now possible to identify biosynthetic genes in bacterial genome sequences and predict the chemical structure of the small molecules. The genome-mining strategy has led to the discovery of numerous biosynthetic gene clusters in genomes of host-associated symbionts[28–30]. A systematic exploration may discover a wealth of natural products produced by the bee gut microbiota.

Here, we systematically investigate the biosynthetic capacity of the bee gut microbiota, including 477 genomes of bacteria from the guts of honey (*A. mellifera*, *A. cerana*) and bumble bees (*Bombus* spp.). The global analysis identifies 744 BGCs encoding small molecules, which covers a broad spectrum of chemical classes. The core gut members show a diverse array of BGCs encoding ribosomally synthesized, post-translationally modified peptides (RiPPs), and almost all of them belong to yet uncharacterized gene cluster families (GCFs). Specifically, we found that the lanthipeptides are enriched in the genomes of *Lactobacillus* species, and an unknown type of RiPP is predominant in *Gilliamella*. Interestingly, functional characterization of the RiPPs from *Gilliamella* revealed potent antibacterial activity against the pathogenic *M. plutonius*. Using a spectrometry-guided genome mining approach, we confirmed that the RiPP-encoding BGC containing two core biosynthetic genes (a cyclodehydratase and a dehydrogenase) possess a 10 amino acid-long core peptide with potential unidentified RiPP motifs.

## Results

### Biosynthetic potential of the bee gut symbionts

To explore the extent of secondary metabolite diversity coded by the bee gut bacteria, we detected biosynthetic gene clusters in 449 genomes of isolates originating from both honey and bumble bee guts and 28 metagenome-assembled genomes from honey and bumble bee guts (Supplementary Data 1)[31]. Using the antiSMASH pipeline, we identified 744 biosynthetic gene clusters for a broad range of small molecule classes (Fig. 1a). The majority of BGCs were categorized as ribosomally synthesized and post-translationally modified peptides (RiPPs) (275), followed by Aryl polyene (122), nonribosomal peptides (NRPs) (97), Terpene (95), and polyketides (PKS) (68) (Supplementary Data 2). RiPPs were the most abundant BGC family (36.9%) in bee gut core bacterial members (Fig. 1a, b). They were mostly identified from the genomes of Gram-positives (83.6%), such as *Lactobacillus* Firm5, *Bombilactobacillus*, and *Bifidobacterium*. In *Gilliamella* strains, NRPS (34.9%) and RiPPs (20.7%) were the most abundant BGC families, while Terpene was mainly encoded by *Snodgrassella* (78.7%), *Commensalibacter* (11.2%), and *Bartonella* (8.7%) species (Fig. 1b, c). Notably, we found that different strains from the same genus exhibited a high extent of diversity in BGC contents (Fig. 1a; Supplementary Fig. 1-5). For example, all *Bombilactobacillus mellis* had cyclic lactone autoinducer genes, which were absent in strains from *Bombilactobacillus mellifer* and those from bumble bees (Supplementary Fig. 4). In addition, the Bacteriocins family from the RiPP-like class were only detected in *Lactobacillus* Firm5 strains from bumble bees (Supplementary Fig. 1). Taken together, our analysis showed the existence of divergent BGCs in the genomes of bacteria isolated from bumble and honeybees, and it showed a high extent of fine-scale strain diversity in the biosynthetic capacity.

### BGC distribution in the gut microbiota of honeybees and bumblebees

To determine the distribution of the identified BGCs across different bee species, we created a BGC map for the systematic identification of biosynthetic clusters identified in bee gut microbiota. First, we merged the 744 BGCs from the bee gut bacteria with 2502 reference BGCs from the Minimum Information about a Biosynthetic Gene cluster (MIBiG) database[32]. We used BiG-SCAPE[33] to generate sequence similarity networks with default "global" mode for all BGCs annotated by antiSMASH from the bee gut bacterial genomes. Considering the weighted combination of Jaccard, adjacency, and domain sequence similarity scores, Pfam composition similarity was used to calculate distances among BGCs. Two rounds of affinity propagation clustering resulted in 178 Gene Cluster Families (GCFs) with distinct core genetic components from 34 Gene Cluster Clans (GCCs) (Fig. 2a; Supplementary Data 3). Remarkably, nearly all of the BGCs annotated from the bee gut dataset belong to undescribed clusters. Only two defined BGCs, paenilamicin (BGC0001033)[34,35] and sevadicin (BGC0000426)[36] from MIBiG fell into three GCFs with 37 BGCs from *P. larvae*, which is a pathogenic bacterium affecting bee larvae[9]. This indicates that the BGCs identified from the bee gut microbiota were distinct from the currently experimentally characterized BGC families. Although each GCC contained gene clusters encoded by different bacteria, almost all GCFs were only represented by bacteria from the same genera, suggesting that the GCFs were unique to different bee gut bacteria (Fig. 2a). For example, T1PKS.hglE-KS was specific to *Apibacter* species, while T3PKS was only detected from *Apilactobacillus kunkeei* (Fig. 2a; Supplementary Data 2). Notably, a large GCF of terpene BGCs was represented by the

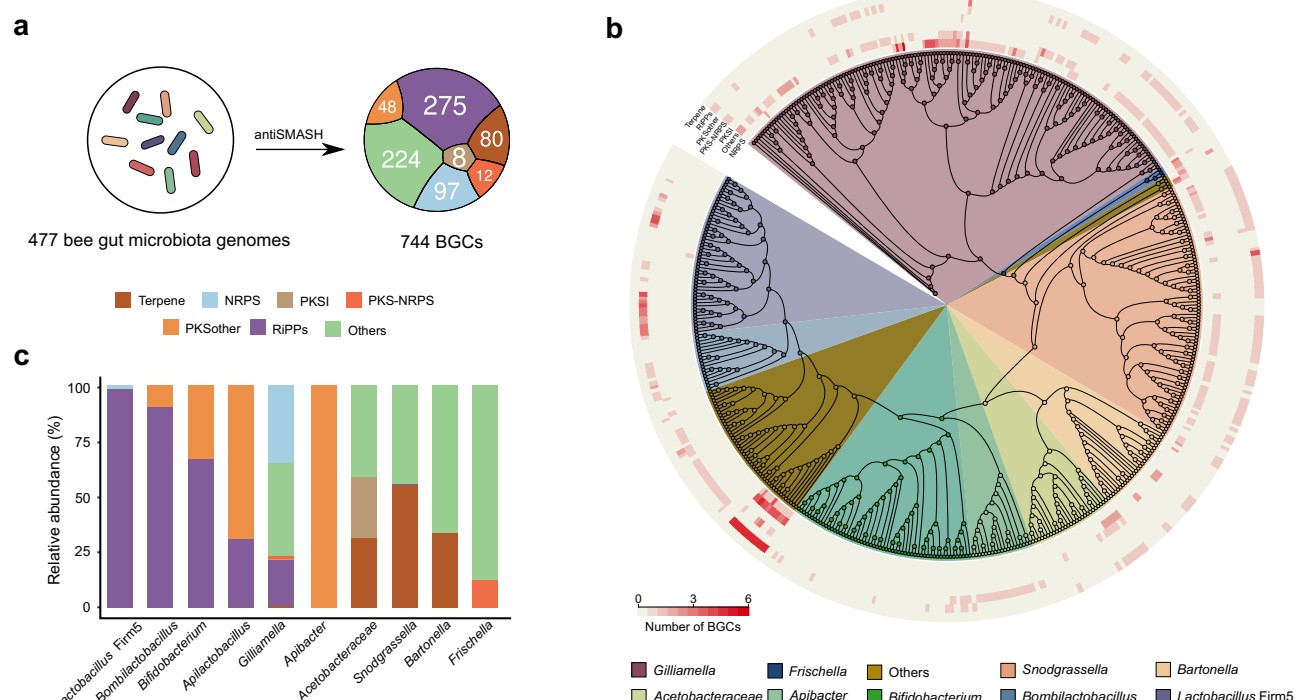

**Fig. 1 | Overview of BGCs predicted in honeybee and bumblebee gut bacteria.**
**a** Overall BGCs were identified from 477 bee gut microbiota genomes. The numbers inside the circle indicate the count of different BGC types. **b** Whole-genome phylogenetic tree of strains from honeybees and bumblebees with the maximum-likelihood algorithm by PhyloPhlAn 3.0 that uses the 400 universal marker genes. Layers surrounding the genomic trees indicate the classes of BGC, and the heatmap shows the numbers of genes belonging to BGCs in each strain, as predicted by antiSMASH. **c** A bar graph showing the relative abundance of BGC types in different bacteria genera from honey and bumble bee gut. Source data are provided as a Source Data file.

pathways from the gut-wall associated symbionts, *Snodgrassella* and *Candidatus* Schmidhempelia.

Considering that BGCs encoding small molecule products were widely distributed in the honey and bumble bee gut symbionts, we next identified the distribution and abundance of the BGCs in honey and bumble bee samples collected from different locations. First, we established a non-redundant database with 152 representative BGCs from the bee gut bacterial genomes. We then used Big-Map[37] to recruit reads from 135 shotgun metagenomes of honeybee individuals (*A. cerana*, *A. mellifera*) sampled from Switzerland[38], Japan[39], and China[17], and of *B. terrestris* individuals from China[31] (Supplementary Data 4, Supplementary Data 5). In general, *A. cerana* and *B. terrestris* samples harbored fewer BGCs than *A. mellifera*, which mirrored the higher microbial genetic diversity in western honeybees[31]. The principal coordinate analysis showed that the BGC profiles were significantly different among the three bee species (Fig. 2b). RiPPs were the most abundant BGC class in the bee guts, while the distribution of the RiPP classes differed among samples from different host species and geographic locations (Fig. 2c, Supplementary Fig. 6a, b).

### Characterization of lanthipeptide from the core microbiota member *Lactobacillus* Firm-5

Lanthipeptides were the most broadly distributed RiPP family throughout bee gut bacterial genomes, mainly encoded by *Lactobacillus* Firm5 (Fig. 2c; Supplementary Data 2). However, the distribution of lanthipeptides showed a high degree of variability among metagenomic samples (Fig. 2c), possibly due to the strain-level diversity of *Lactobacillus* Firm5 in different bee individuals. Lanthipeptides are divided into five classes depending on the characteristics of their biosynthetic enzymes[40]. In the genomes of *Lactobacillus* Firm5, we identified four classes of lanthipeptides (Lanthipeptide I-IV), while the

distribution showed apparent intra- and inter-specific variations (Fig. 3a). While Most strains of Frim5 from bumble bees (*L. bohemicus*, *L. bombicola*) encode Lanthipeptides III, *L. helsingborgensis* strains ESL0262, ESL0183 and wkB8 from the honeybee possessed Class I and IV lanthipeptides. Only few strains of *L. panisapium* from *A. cerana* and of *L. melliventris* had genes encoding lanthipeptides. Lanthipeptide biosynthesis starts with the ribosomal synthesis of the precursor peptide that is matured by enzymes encoded by core and additional biosynthetic genes, and the matured peptide is secreted via transporters[41]. We explored the arrangements of BGCs of lanthipeptides in genomes of Firm5 strains (Fig. 3b). While the arrangements of the BGC were different in Firm5 strains, the core biosynthetic genes showed synteny within each class.

We found that Lanthipeptide I from *L. melliventris* Hma8 had a conserved combination of aminoacyl-tRNA-dependent dehydratase (LanB) and cyclase (LanC)[42]. The LanM with N-terminal dehydratase and C-terminal cyclase domains[43] was detected from Lanthipeptide II of *L. bombicola* ESL0230 (Fig. 3c). Moreover, in the Lanthipeptide II of strain ESL0230, three precursor peptides were predicted by antiSMASH (Fig. 3b). The precursor peptides are typically composed of the N-terminal leader peptide and the C-terminal core peptide[44]. We identified that the core peptide of ESL0230 possessed Ser, Thr, and Cys amino acid residues required for the formation of Lan and MeLan thioether rings (Fig. 3d). Typically, Ser and Thr are dehydrated to 2,3-didehydroalanine (Dha) and (Z)−2,3-didehydrobutyrine (Dhb), respectively (Supplementary Fig. 7a, b). Correspondingly, Lanthipeptide II-specific bifunctional proteases (LanT$_p$) responsible for the leader removal and the export of final products were also identified in ESL0230 (Fig. 3b).

Comparatively, the core biosynthetic gene of Lanthipeptide III (LanKC) and IV (LanL) contains three typical domains: an N-terminal lyase, a central kinase, and a C-terminal cyclase (Fig. 3C)[45]. An

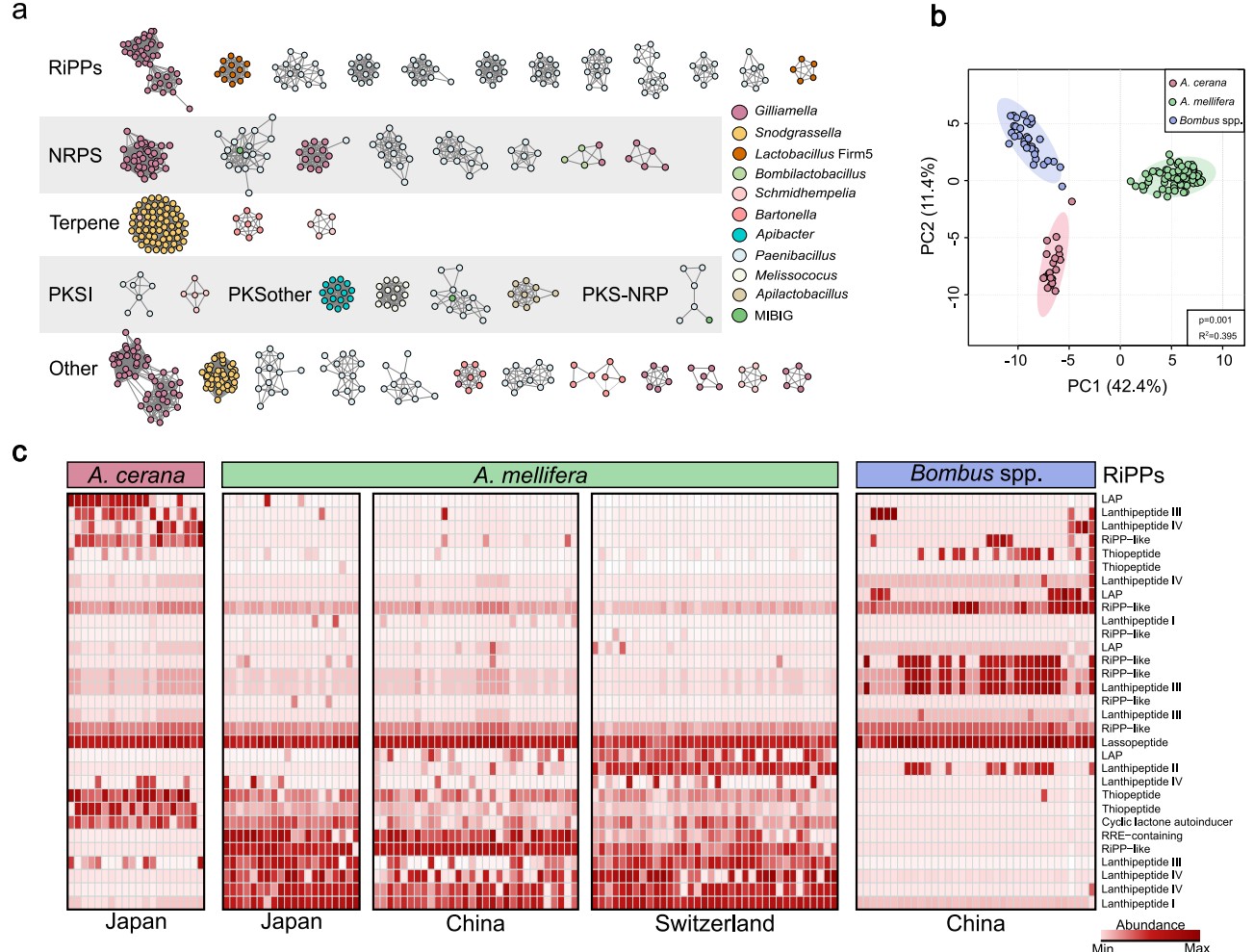

**Fig. 2 | BGCs of the gut microbiota distinguish between honey and bumble bee species. a** Network depicting the similarity between putative BGCs identified from bacterial genomes of the bee gut microbiota and a curated collection of functionally characterized BGCs (MIBiG[32], 2502 sequences). The BGCs were clustered into GCFs and separated into seven classes with BiG-SCAPE[33]. Only GCFs containing at least five BGCs were shown here. Nodes in the network represent BGCs, and the edges connect BGCs with a similarity ≥0.7, defined by BiG-SCAPE.

**b** PCA analysis shows the BGC distribution in the gut of *A. cerana*, *A. mellifera*, and *B. terrestris* individuals. Group differences were tested by one-way permutational multivariate ANOVA (PERMANOVA). **c** The distribution of RiPPs across bumblebee (*B. terrestris*) and honeybee (*A. cerana*, *A. mellifera*) gut metagenomes. Each column represents one bee gut sample. The relative abundance of BGCs (RPKM) was log-transformed and normalized by the median. Source data are provided as a Source Data file.

alignment revealed that the lyase domains of Firm5 shared the conserved residues with exemplary Lanthipeptide III/IV enzymes, which are required for the catalysis (Fig. 3c). Similarly, conserved residues were detected from structural motifs, including *helix C* motif (Glu), P-loop (Gly, Gly), catalytic loop (Asp, Asn), and DFG (Asp) of the central kinase of Firm5 strains (Supplementary Fig. 8a, b). Although the zinc-binding residues were conserved across Lanthipeptide I, II, and IV, they were absent from the cyclase domain of LanKC (Supplementary Fig. 8a, b). These results illustrated that the core biosynthetic regions from bee gut *Lactobacillus* possessed conserved structural features with characterized lanthipeptides important to the activity.

**The inhibitory effect of RiPPs-like peptides from *Gilliamella* on pathogenic *Melissococcus***

We noticed that *Gilliamella*, the Gram-negative core bee gut member, also encoded a distinct GCF of RiPPs, which are annotated as potential thiopeptides by antiSMASH (Fig. 1a, b). These BGCs did not show connections with any reference clusters from the MIBiG database[32], suggesting previously unexplored biosynthetic capacity (Fig. 2a). Interestingly, the RiPPs were present in almost all strains of *G. apis* and some strains of another undefined species cluster *Gilliamella* sp.

(Supplementary Fig. 5). However, this BGC was absolutely absent in *G. apicola* strains (Fig. 4a).

RiPPs from host-associated symbionts often exhibit broad-spectrum inhibitory effects on pathogenic bacteria[46]. We wondered whether the specific RiPPs from the bee gut *G. apis* could be active against honeybee pathogens. Here, we used *M. plutonius*[47], a Gram-positive etiological agent of European foulbrood, to test the antipathogenic effect of the RiPPs. Considering that not all *Gilliamella* strains possess the BGCs of the RiPPs, we chose two *G. apis* strains (B14384H2, W8126) encoding the RiPPs and two *G. apicola* strains (W8136, G14384G12) that do not encode the RiPPs. We tested the effect of the cell-free supernatant of these strains on the growth of *M. plutonius* in a disc-diffusion assay in vitro. Interestingly, only the supernatants of *G. apis* B14384H2 and W8126 inhibited the growth of *M. plutonius*, while no antibacterial activity was observed for the supernatants of *G. apicola* (Fig. 4b). In addition, we also tested the activity of the supernatants of *Lactobacillus* Firm5 encoding lanthipeptide BGCs, but no inhibition was observed (Supplementary Fig. 9). To further evaluate whether *G. apis* strains could inhibit the invasion of *M. plutonius* in vivo, we colonized MF honeybees with different *Gilliamella* strains (Fig. 4c, d). After 7 days of colonization, all these stains grew to

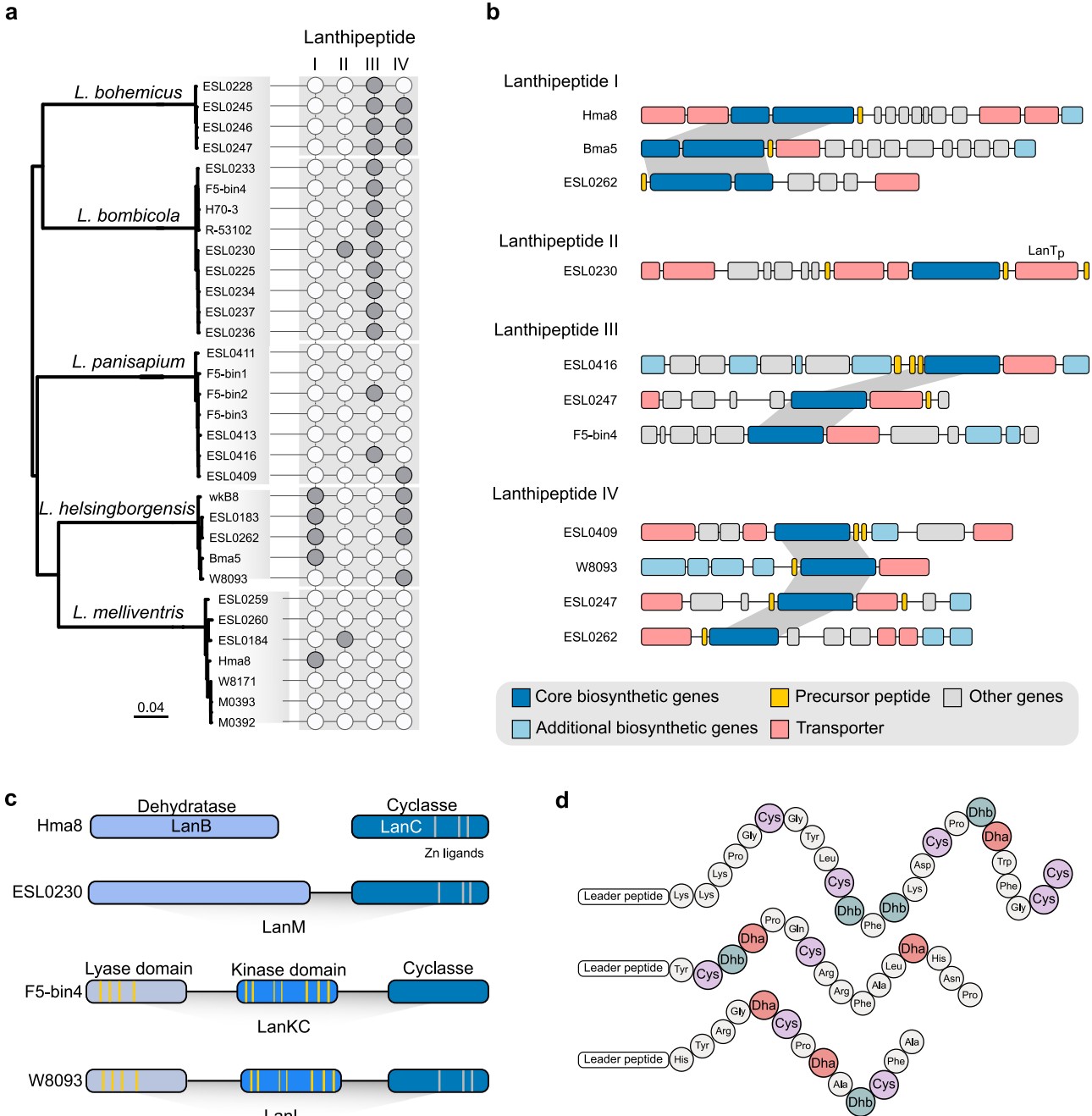

**Fig. 3 | Phylogenetic diversity of lanthipeptide BGCs code by *Lactobacillus* Firm5. a** Phylogenetic relationships of the *Lactobacillus* Firm5 coding lanthipeptide BGCs. Cladograms are maximum-likelihood trees inferred by GTDB-tk based on the amino acid sequences of bacterial marker genes. Layers surrounding the genomic trees indicate four classes of lanthipeptides in *Lactobacillus* Firm5 genomes, as predicted by antiSMASH[73]. A full tree is shown in Supplementary Fig. 1. Gray circles indicate gene presence, and empty circles indicate gene absence. **b** Syntenic loci of core biosynthetic genes, precursor peptide, additional biosynthetic genes, and transporter genes for lanthipeptide in *Lactobacillus* Firm5 strains. Homologous genes are connected by gray bars. **c** Schematic representation of the four classes of lanthipeptide synthetases, highlighting conserved motifs. LanB, lanthipeptide dehydratase; LanC, lanthipeptide cyclase; LanM, class II lanthipeptide synthetase; LanKC, class III synthetase; LanL, class IV lanthipeptide synthetase. The yellow bars in the lyase and kinase domains from LanKC and LanL indicate conserved regions that are important for catalytic activity. **d** Predicted three precursor peptides dehydration reactions during the LanM of strain ESL0230 deduced from antiSMASH 5.0[73].

~2.0 × 10⁷ CFU/bee (Supplementary Fig. 10). Then, each bee was orally infected with an exact amount of *M. plutonius* individually (10⁶ cells per bee). On Day 14, we found that the absolute abundance of *M. plutonius* was decreased in the gut of mono-colonized bees with *G. apis* strains B14384H2 and W8126, while they did not show a significant difference with bees colonized by *G. apicola* strains W8136 and B14384G12.

We next explored the potential active ribosomally synthesized and post-translationally modified peptides against *M. plutonius*. The

antiSMASH analysis predicted that the RiPP gene cluster from *G. apis* strains contained two core biosynthetic genes, a cyclodehydratase, a radical SAM methyltransferase, and two nearby transport-related genes, which are essential to the post-translational modifications of RiPPs (Fig. 4e). Although antiSMASH predicted the RiPP-encoding BGC, it is limited to deciphering the specific BGC organization of RiPPs with unknown post-translational modifications. Therefore, we used a spectrometry-guided genome mining method to confirm the ORF

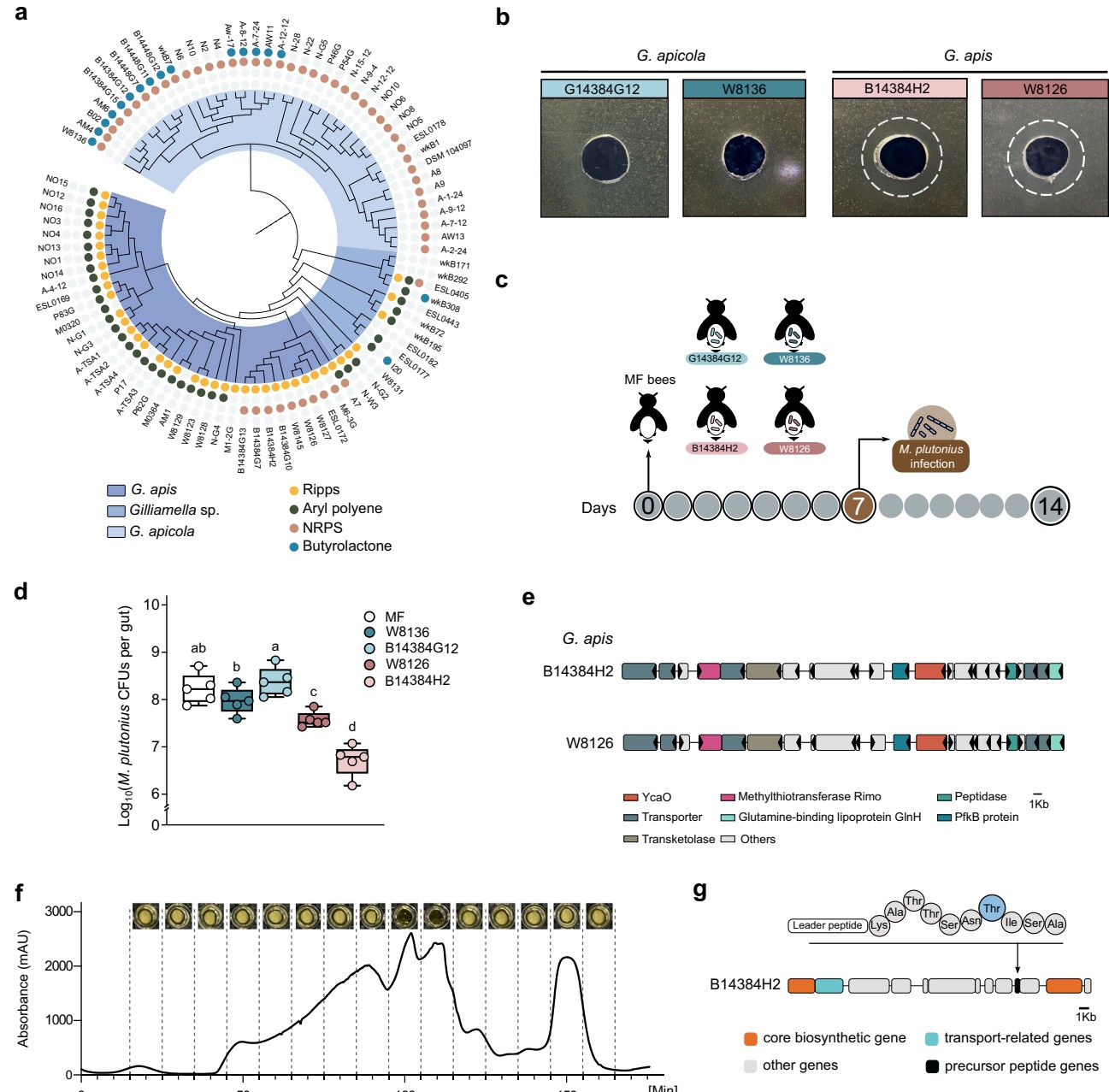

**Fig. 4 | *Gilliamella* strains protect against *M. plutonius* infection through the secreted RiPPs in honeybee gut. a** Full phylogenetic tree of the *Gilliamella* genus based on the whole-genome sequences using the maximum-likelihood algorithm. The inner layer surrounding the genomic tree designates taxonomic annotations, while the remaining layers depict the classes of BGCs in the genome. A full tree is shown in Supplementary Fig. 5. **b** The cell-free supernatant of *G. apis* B14384-H2, W8126 inhibits the growth of *M. plutonius* in vitro. **c** Schematic of the in vivo experiments. MF bees were colonized with *G. apis* B14384-H2, W8126, *G. apicola* W8136, and B14384G12 for 7 days and then orally infected with *M. plutonius* (10⁶ bacterial cells per bee). **d** Absolute abundance of *M. plutonius* of different treatment groups 7 days post-inoculation with *M. plutonius*. The overall significance was first assessed by one-sided ANOVA ($p = 0.001$). Duncan's test

was used for multiple comparisons. The letters above each bar show statistical differences among the treatments ($p < 0.05$) ($n = 5$ bees for both groups). Box boundaries are the 25th and 75th percentiles, the horizontal line across the box is the median, and the whiskers indicate the minimum and maximum values. **e** Organization of the *G. apis* B14384H2 and W8126 strains RiPPs biosynthesis cluster. The proposed biosynthetic genes and surrounding regions are displayed. **f** Base peak chromatograms of extracts from *G. apis* B14384H2 with images of the antibacterial activities by micro-plate assay showing the bioactivity of 95–115 min fractions correlating to the portion of the chromatogram against *M. plutonius*. **g** The precursor peptide of the RiPPs was identified by MetaMiner[48]. Post-translationally modified amino acids are shown in blue. Source data are provided as a Source Data file.

encoding the precursor peptide within the BGC. To verify the active RiPPs produced by *G. apis*, we purified the supernatant extracts from bacterial cultures of strain B14384H2 using size-exclusion chromatography (Fig. 4f). We tested the inhibitory effect of each separate fraction of the extracts against *M. plutonius*. Interestingly, only two consecutive fractions with a retention time of 95–115 min inhibited the

growth of *M. plutonius* (Fig. 4f). We then analyzed the bioactive supernatant extracts by liquid chromatography-mass spectrometry and generated the tandem mass spectra. Then, we constructed a database of putative RiPP structures based on the genome of strain B14384H2 using the full-ORF model of MetaMiner, which identified all short ORFs within the BGCs and considered the modifications for the

putative precursor peptide[48]. By matching the LC-MS/MS spectra of the active fraction against the RiPP structure database, we found a 132-bp long ORF neighboring the *pfkB* gene with a significant peptide-spectrum match (*p*-value = 8.1 × 10⁻¹¹) at the conservative FDR (0%) (Supplementary Table 1). This ORF is predicted to encode a 33-aa long leader peptide and a 10-aa long core peptide showing potential post-translational modifications (KATTSNT-18ISA). Thus, we demonstrated that *Gilliamella* strains with specific BGCs produce an unkown type of RiPP, which is active against the pathogenic *M. plutonius*.

## Discussion

In this study, we systematically identified the distribution of BGCs in the genomes of gut bacteria isolated from honeybees and bumblebees. Bee gut symbionts exhibit great potential for the synthesis of bioactive natural products. Most of these BGCs are distantly related to the characterized products from the MIBig database and showed a high degree of diversity at the strain level. The identified 744 BGCs from 477 bacteria genomes were categorized as RiPPs, Aryl polyene, NRPs, Terpene, and PKS. Furthermore, we found significant variations in the distribution and abundance of RiPPs among different bee species and even within the same species from different geographical locations. Particularly, the distribution of lanthipeptides exhibits high variation among metagenomic samples, probably due to the fine-scale strain diversity within *Lactobacillus* Firm5 species in the biosynthetic capacity. In addition, we characterized a RiPP antibiotic from *G. apis* showing potent activity against *M. plutonius*, highlighting the potential of these unknown RiPPs as promising candidates for combating infectious diseases.

Many studies have revealed species-specific, uncharacterized small-molecule-encoding BGCs in the gut microbiome of humans[21], swine[49], and ants[50]. Most of the BGCs are always enriched in the members of gut-associated bacteria, such as *Bacteroides*, *Parabacteroides*, and *Ruminococcus*, and the chemical classes from the gut microbiota exhibit notable differences compared to non-gut-associated bacteria[21]. The honey and bumble bee gut comprises host-specific bacterial phylotypes, which have evolved for ~80 million y with their hosts[18]. In the first step toward exploring the biosynthetic capacity of the honeybee gut microbiome, we found core bacterial members from the bee gut have multiple BGCs in each genome, indicating that the bee gut is a rich resource for secondary metabolites. The global analysis revealed a high diversity of predicted RiPPs being the most abundant BGC class in the bee gut and other natural products, which parallels the profiles of other gut communities[21,30]. Intriguingly, almost all identified BGCs from the bee gut microbiota did not overlap with the references database, and the dereplicated GCFs were not represented in MIBiG[32], which is one of the most extensive databases of experimentally validated BGCs. These indicated a discovery of previously undescribed bioactive pathways from the bee gut-specific bacteria. Furthermore, although the genome size of the bacteria restricted to the bee gut ecosystem is relatively small, ranging from 1.5 to 3 Mb, they encode, on average, ~2 large BGCs per genome. This indicated that the encoded small-molecule products must play important ecological roles in specific phenotypes potentially by mediating microbe-host interactions[21].

Among these, aryl polyenes are a group of natural products widespread in taxonomically distant bacteria[51]. They are structurally similar to the carotenoids, and this class of pigments can protect the bacteria from reactive oxygen species[52]. Previous investigations showed that the deletion mutants of the aryl polyene biosynthesis pathways in *F. perrara* exhibited decreased colonization levels in the pylorus. We found that the aryl polyene BGCs are also enriched in *Gilliamella* and *Snodgrassella* strains from the honeybee gut while less represented in *Bombus*-associated strains. Notably, these two bee gut members always form contiguous biofilms at the inner gut wall of the bee ileum[18]. Furthermore, we found that another class of

secondary metabolite, terpene, is specifically present in almost all strains of *Snodgrassella*. Terpene is also an extensive compound in bacteria, which provides relevant protection under oxidative stress conditions[53]. It is noteworthy that the previous Tn-seq analysis showed that several genes belonging to the terpene BGC of strain wkB2 are highly beneficial for colonization (SALWKB2_RS05205, SALWKB2_RS05220, SALWKB2_RS05275) or essential for growth (SALWKB2_RS05210, SALWKB2_RS05235)[54]. These findings suggest that these natural products may act as antioxidant molecules preventing the stress caused by reactive species from the host, which is implicated in mediating host-symbiont interactions in complex environments.

RiPP natural products, which are usually prevalent among gut-associated bacteria[21], are also found to be the most represented chemical class in the bee gut community. RiPPs are divided into many subclasses[55], and we identified five members of defined RiPP subclasses from the bee gut bacteria, including cyclic-lactone-autoinducer, lanthipeptide, LAP, ranthipeptide, and thiopeptide. However, the distribution of the RiPP classes varied significantly in different bee species and even the same bee species among geographic locations (Fig. 2c). For example, the lantibiotics are the most common RiPPs identified from the bacteria isolated from bee guts, and they are divided into five subclasses according to their structure and biosynthetic machinery[40]. Lanthipeptide III was enriched in the bumblebee gut community, while Lanthipeptide I and IV are broadly distributed in *A. mellifera* samples. This variation may be attributed to the inter- and intra-species diversity in the BGC profile of bee gut *Lactobacillus*, which is the main carrier of the lanthipeptides. Most characterized class I and II lanthipeptides are known for their antibiotic properties. These lantibiotics predominantly target the cell wall precursor lipid II and inhibit cell wall biosynthesis effectively. Previous studies on *A. kunkeei*, a non-core member of the honeybee microbiota, revealed the presence of a class I lanthipeptide, kunkecin A, which potentially inhibits the growth of other gut bacteria[26]. In our study, Lanthipeptide I was identified in the genome of *L. helsingborgensis* and *L. melliventris* strains, which may also be important in the host's defense against pathogens. Moreover, our analysis identified a more diverse distribution of Lanthipeptides III and IV in the genomes of *Lactobacillus* Firm5. Compared to Lathipeptide I and II, class III and IV lanthipeptides often lack antibacterial activities[56]. The antiSMASH analysis showed that the Lanthipeptides III of *Lactobacillus* Firm5 possessed the conserved residues within the lyase, kinase, and cyclase enzymes, and they are closely related to those from *Lactobacillus delbrueckii* and *Bacillus* species. Recently, a series of Lanthipeptide III derived from the gut-associated *Bacillus amyloliquefaciens* exhibited a narrow antimicrobial spectrum against a limited set of species closely related to the producer[57]. In the honeybee gut, phylogenetically close species of *Lactobacillus* Firm5 stably coexist while showing variable abundance among individuals[17]. So far, nutrient partitioning was hypothesized to partially explain the coexistence of the species[27]. Further evaluation of the bioactivity of the lanthipeptides derived from the bee gut *Lactobacillus* may illustrate their vital roles in mediating interactions among closely related species, which is implicated in gut homeostasis[27].

Additionally, RiPPs are a promising source of alternative antimicrobials, especially as antibiotic resistance has become a growing crisis[58]. Specifically, thiopeptides are a large group of structurally complex natural products that always show strong antibacterial activity against Gram-positives[59]. Here, we investigated that strains from *G. apis*, but not *G. apicola*, inhibit the growth of *M. plutonius*, while they are both predominant *Gilliamella* species in the honeybee gut[17,39]. *G. apis* strains harbor the BGCs of potential thiopeptide, which react as protein synthesis inhibitors by binding to the bacterial ribosome. Since the binding sites of thiopeptide are distinct from used antibiotics, they may overcome the existing antibiotic resistance in pathogens[60]. For example, thiazomycin show potent activity against

the multidrug-resistant *Mycobacterium tuberculosis*[61], and nosiheptide effectively inhibits the methicillin-resistant *Staphylococcus aureus* and vancomycin-resistant *Enterococcus*[62]. The antiSMASH analysis identified the BGC of *G. apis* containing a set of genes, including the core biosynthetic genes for thiopeptide-like BGC, such as the azoline-forming YcaO cyclodehydrating Cys, Ser, and Thr residues, radical SAM-dependent methyltransferase, ABC transporters, and the peptidases[63]. However, this BGC from *Gilliamella* differs significantly from the currently known RiPPs in the database. Future investigations on the post-translational modification and maturation process are needed to solve the structure of the identified RiPP from bee gut symbionts.

EFB is purely a disease of the digestive tract of the larvae[64], while the first step in EFB infection is an asymptomatic colonization of larvae due to the food transmission by adult nurse bees[65,66]. Additionally, adult workers removing infected larvae may transfer pathogens to healthy broods within colonies[67] and even between colonies through robbing and drifting[68]. Erban et al.[12] showed that *M. plutonius* reached ~$10^6$ CFU/bee in the adult bees from colonies with clinical symptoms, which is 75-fold-higher than those from asymptomatic colonies. Our results showed that gnotobiotic bees mono-colonized with *G. apis* stain B14384H2 can achieve ~100-fold reduction of *M. plutonius* (on average, from $2.12 \times 10^8$ to $6.22 \times 10^6$, $p = 0.0079$). However, future studies on the interactions between adult bees and larvae are needed to investigate the role of potential bioactive molecules in the infection control. Honeybee gut bacteria adapting to the gut environment and competing against non-native microorganisms may drive the selection of BGCs that produce highly efficacious natural products. Our study highlights the distribution of BGCs with uncharacterized functions in the bee gut microbiome and provides insights into the potential therapeutic agents for bee diseases.

## Methods
### Phylogenomic analysis
We compiled a dataset comprising 449 genomes of isolates originating and 28 metagenome-assembled genomes from honey and bumble bee guts (Supplementary Data 1). To construct phylogenetic trees for all bacterial genomes from bee gut, we used PhyloPhlAn 3.0 employing 400 universal marker genes under the 'diversity low' parameter[69], and the maximum likelihood method was used. To create the whole-genome tree for the core bacterial genera, i.e., *Lactobacillus* Firm5, *Bombilactobacillus*, *Gilliamella*, *Snodgrassella*, and *Bifidobacterium*, we used Roary version 3.12.0[70] with the parameter '-blastp 75' to identify core single-copy genes shared among all strains. The resulting alignments of nucleotide sequences were concatenated, and the maximum-likelihood trees were inferred using FastTree version 2.1.10[71] with a generalized time-reversible (GTR) model. We utilized the iTOL web-based software[72] for the visualization of these phylogenetic trees.

### Identification of BGCs from isolate genomes and shotgun metagenomes of honeybees and bumblebees
Genome sequences of 477 bacterial strains from the guts of honeybees and bumblebees were explored for biosynthetic gene clusters (BGCs) using the antiSMASH 5.0 under the Fast run settings, which ran the core detection modules and all fast cluster-specific analysis steps[73]. This identified a total of 744 BGCs from the bacterial genomes with known categories (Supplementary Data 2). Each predicted BGC was manually inspected for completeness to exclude the clusters truncated on the edge of the contigs. Next, we consolidated and passed all putative BGCs through the BiG-SCAPE package to explore the interactive BGC sequence similarity networks representing the differences in modes of evolution between BGC classes[33]. The networks were computed using the 'mix' and 'mibig' options to include the BGCs from the MIBiG database[32] in the network. We also used the '--include singletons' option to visualize singleton BGCs in the network. Networks

were tested with multiple raw distance cutoff values (0.1–1.0), and the networks computed using a 0.7 cutoff were chosen. The resulting sequence similarity matrices were then visualized in Cytoscape v.3.7.2.

To profile the prevalence of the metabolic gene clusters across different bee gut samples, we analyzed 135 publicly available shotgun metagenomes of bumble and honeybees[17,31,38,39] using the BiG-MAP pipeline[37]. First, we established a bee gut bacteria BGC database using the 'BiG-MAP.family.py' module, which performed redundancy filtering on all BGCs annotated from the isolate genomes as described above and selected representative gene clusters for the mapping process. Then, the reads from 141 bee gut metagenomes were mapped to the representative gene clusters using the 'BiG-MAP.map.py' module. The 'BiG-MAP.analyze.py' was used to normalize the counts for sparsity and sequencing depth, and the abundance of different BGC clusters was calculated.

### Antibacterial assay of the honeybee gut symbionts against *M. plutonius*
Bee gut bacterial strains, *G. apis* B14384-H2 and W8126, *G. apicola* W8136, and G14384-G12, were isolated from the guts of *A. mellifera* and deposited in our lab[17]. *Gilliamella* strains were routinely grown on Heart Infusion Agar (HIA) (Oxiod, Hampshire, UK) supplemented with 5% (vol/vol) sterile sheep blood (Solarbio, Beijing, China). To test the antibacterial activity, cell suspensions of bee gut strains were prepared by collecting colonies from the plates and diluting them to a final concentration of ~$10^8$ CFU/mL in 1×PBS. The cell-free supernatant was obtained by centrifuging bacterial suspensions at 5000 × *g* for 5 min, followed by filtration through a 0.22-μm pore size syringe filter (Minisart 16532-K, Sartorius, Göttingen, Germany). *M. plutonius* ATCC 35311 was grown on KSBHI medium at 35°C. KSBHI medium was composed of HIA medium supplemented with 20.4 g/L of $KH_2PO_4$ and 10 g/L of soluble starch. Colonies were collected from the plates and diluted to $10^6$ CFU/mL using 1×PBS. Then, 100 μL of the *M. plutonius* liquid culture was spread over the surface of the KSBHI agar plates. A well with a 10-mm diameter size was made in the middle of the agar plates (Fig. 4b), and 100 μL of the supernatants of *Gilliamella* strains were put into the well. The plates were incubated at 35°C and under 5% $CO_2$ for 48 h.

### Bees mono-colonized with gut symbionts challenged with *M. plutonius*
MF bees were obtained as described by Zheng et al.[19]. Briefly, late-stage pupae were removed manually from brood frames and placed in sterile plastic bins. The pupae emerged in an incubator at 35°C, with a humidity of 50%. Newly emerged MF bees (Day 0) were kept in axenic cup cages with sterilized sucrose (50%, vol/vol) syrup for 24 h. For each mono-colonization setup, 20–25 MF bees were placed into one cup cage, and the bees were fed on the bacterial culture suspensions for 24 h. For the MF group, 1 mL of 1×PBS was mixed with 1 mL of sucrose solution and 0.3 g sterilized pollen. For the other group, glycerol stock of bee gut strains was resuspended in 1 mL 1×PBS at a final concentration of ~$10^8$ CFU/mL and then mixed with 1 mL sterilized sucrose solution. All bees were kept in an incubator (35°C, RH 50%) until Day 7.

To precisely control the infection amount of *M. plutonius* cells, bees were individually inoculated with *M. plutonius* by oral feeding. Cell suspensions of *M. plutonius* were prepared by collecting colonies from plates into 20% sucrose in 1×PBS. Each bee individual was starved for 3 h and was fed with 5 μL of the cell suspension. Inoculum levels of *M. plutonius* were ascertained by enumerating CFUs from plated serial dilutions of the cell suspension. Each bee individual was fed exactly 1 × $10^6$ CFUs of *M. plutonius*. After 7 days, we determined the loads of *M. plutonius* in gut samples by qPCR as previously described[19]. *M. plutonius*-specific primer sets were used (F-TCAACCGGGGAGGGTCATT, R-AGCCTCAGCGTCAGTTACAG).

## Purification of thiopeptide by gel column chromatography

*G. apis* B14384-H2 was cultured in Heart Infusion broth (Oxiod, Hampshire, UK) medium at 35°C for 5 days. The liquid culture was centrifuged at $7000 \times g$ for 15 min in a high-speed refrigerated centrifuge (Thermo Scientific, Waltham, MA, USA) to remove bacterial cells. The remaining liquid was filtered through a membrane with 0.22-μm pore size to yield a cell-free supernatant. The obtained supernatants were then purified using a Superdex™ 30 Increase 10/300 GL size exclusion chromatography column on a pure protein separation and purification platform (GE Healthcare, Marlborough, MA, USA). The following purification platform settings were used for data acquisition: equilibrium volume of 2 column volumes (CV); elution at pH 5.2; elution volume of 1.5 CV; UV 280 nm; flow rate of 0.3 mL/min. 0.5 mL of each substance under the corresponding eluted peak was collected into a 15 mL tube. Next, the collected supernatants were diluted two-fold in KSBHI and mixed with an equal volume of the cell suspensions of *M. plutonius* in KSBHI ($10^6$ CFUs/mL). The mixture was cultured in a clear UV-sterilized 96-well plate at 35°C for 48 h. Then the fractions showing the highest antibacterial activity were then solidified by freeze drying in a lyophilizer and stored at 4°C until further analysis.

## Identification and molecular mass determination

The molecular mass of the extracted peptides was determined by Nano LC-MS/MS analysis. Briefly, eluted samples were reduced with 10 mM DL-dithiothreitol (Macklin, Shanghai, China) at 56 °C for 1 h and then alkylated with 20 mM iodoacetamide (Macklin) at room temperature in the dark for 1 h. Peptide was then subjected to lyophilization and resuspended in 10 μL of 0.1% formic acid (Sigma-Aldrich, St. Louis, MO, USA) before chromatography analysis. Analysis of peptides was conducted by nano LC-MS/MS in an Ultimate 3000 system (Thermo Scientific) coupled with an Orbitrap Elite™ Hybrid Ion Trap-Orbitrap mass spectrometer (Thermo Scientific) with an electrospray nanospray source. An in-house built reverse-phase nanocolumn of 150 μm × 15 cm, packed with ReproSil-Pur C18-AQ 1.9 μm resin (100 Å; Dr. Maisch GmbH, Ammerbuch-Entringen, Germany) was used. A linear gradient elution of acetonitrile (Sigma-Aldrich, St. Louis, MO, USA) was used. Mobile phase A consisted of 0.1% formic acid in ultrapure water, and mobile phase B comprised 0.1% formic acid in acetonitrile. The following procedure was used: from 6% to 9% B for 5 min; from 9% to 14% B for 15 min; from 14% to 30% B for 30 min; from 30% to 40% B for 8 min; and from 40% to 95% B for 2 min, with flow rate at 0.6 μL/min. 5 μL of samples was loaded into the system. Mass spectrometry measurements were performed in a data-dependent scan controlled by Xcalibur 2.1.2 software in Orbitrap (at a spray voltage of 2.2 kV and a capillary temperature of 270 °C). Mass spectrometry analyses were performed with a single full scan (MS) using the following parameters: 100–1500 m/z range; 60,000 resolution at 400 *m/z*; then, 10 data-dependent scans (MS/MS) with 27% normalized collision energy were performed using the following parameters: production ion scanning starting at 100 *m/z*; 1500 minimal signal required; 3.00 isolation width; 40 normalized collision energy, 6 default charge state; 0.25 activation Q; 30 activation time, 50–1500 MS precursor m/z range. The mass spectrum signal was obtained after the MS scan.

Then, we used a metabologenomic pipeline, MetaMiner[48], to integrate the tandem mass spectra and genomic data to identify the RiPP-encoding BGC in *G. apis*. The whole genome of strain B14383H2 (GCA_016101655) and the spectrum file of the active extract fraction were uploaded to the online platform (http://gnps.ucsd.edu). A database of putative RiPP precursor peptides of *G. apis* was constructed with the 'all-ORF' strategy, which detects all putative modification enzymes using HMMer to discover RiPP sequences not similar to any known RiPPs. Then, using Dereplicator[74], the LC-MS/MS spectra of the active fraction were searched against the decoy RiPP database with the following settings: Parent Mass Tolerance = ± 0.02 Da, Product Ion Tolerance = ± 0.02 Da, Product Class = 'ALL'.

## Statistics and reproducibility

*M. plutonius* infection level (qPCR tests) among different groups was detected by one-sided ANOVA and Duncan's test. The exact value of n representing the number of groups in the experiments described was indicated in the figure legends. Any additional biological replicates are described within the Methods and the Results. No statistical method was used to predetermine sample size. No data were excluded from the analyses. The experiments were not randomized. The Investigators were not blinded to allocation during experiments and outcome assessment.

## Reporting summary

Further information on research design is available in the Nature Portfolio Reporting Summary linked to this article.

## Data availability

All data generated in this study are provided within the manuscript files. The accession numbers of all raw data of the metagenomes and the genomes of isolated honeybee gut bacteria are listed in Supplementary Data 5. Source data are provided with this paper.

## Code availability

The list of analysis software and all scripts generated for analysis have been deposited on GitHub at: https://github.com/HaoyuLang/Bee_micro_BGC.git.

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

## Acknowledgements

This work was supported by the National Key R&D Program of China (Grant No. 2019YFA0906500) and the National Natural Science Foundation of China Project 32170495 to H.Z.

## Author contributions

H.Z. supervised the study; H.Z. X.H. and H.L. designed the study; H.L., Y.L., H.D. and W.Z. collected samples and performed the pathogen infection experiments; H.L., Y.L. and H.D. generated data and performed the data analyses with contributions from Y.L., H.D. and H.L. prepared the manuscript.

## Competing interests

The authors declare no competing interests.
