## [Peer Review File · Nature Communications]

REVIEWER COMMENTS

Reviewer #1 (Remarks to the Author):

The paper by Lang et al. constitutes a well-executed piece of research on the biosynthetic diversity of honeybee gut microbiota, culminating in the identification of interesting RiPP natural products. The manuscript is well-written and accompanied by beautiful figures. Generally, I am very positive about the work, but I do have a number of feedback points that the authors could take along in their revisions:

- A description of methods for the whole-genome phylogeny is missing. Which model was used, which markers, which tree construction method?
- For the metagenome data, a list of SRA accessions should be provided so that it is clear which samples were used and to which data columns in the SI tables they correspond.
- Line 114-128: various statements are provided that remain vague ('mostly', 'mainly', etc.); please provide actual numbers wherever possible (also elsewhere in the manuscript).
- Figure 1b: please use the same order of the genera in the color legend as the order in the tree
- Line 145-149: please indicate whether global or glocal mode was used
- Line 153: how many BGCs had matches to MIBiG (again, please be quantitative) and which BGCs were they? Are they related to compounds that are expected to be found in this microbiome?
- For the novel characterized RiPP, it would be nice if the BGC were deposited into MIBiG and the structure elucidated (or potentially inferred from genome + MS/MS data, if possible). Is it predicted to be cyclic?
- Figure 2c: the color scale has two gradients (red to white, white to blue), which I think is more logical for cases in which data can also be negative. I would recommend a single gradient.
- Figure 3: were there any signs of class V lanthipeptides (not detectable with antiSMASH, I think, but could be found using BLAST)?
- Line 304-305: can the peptide-spectrum match be visualized perhaps, and added as an SI Figure?
- Line 462: why was antiSMASH 5.0 used? Historical reasons?
-

Spelling/grammar/style:

- Line 20: 'the' -> 'a' (please also check usage of articles throughout)
- Line 80: the BGCs do not encode compounds, they encode their biosynthetic enzymes
- Line 86: 'The genome mining algorithm': which algorithm?
- Line 94: remove 'the'
- Line 100: 'the' -> 'a'

Reviewer #2 (Remarks to the Author):

In this study, Lang et al. investigated the bee gut microbiota for small molecule biosynthetic gene clusters (BGCs) and identified numerous novel ones, including post-translationally modified peptides and lanthipeptides. They also observed significant variations in BGC distribution among bee species and geographic locations, reflecting strain-level differences within the bee gut microbiota. Additionally, they demonstrated that specific strains of *Gilliamella apis* possess a thiopeptide-like BGC, which exhibited antibacterial activity against the pathogenic bacterium *Melissococcus plutonius* in specific in vitro and in vivo experiments. They also employed a spectrometry-guided genome mining approach to discover the RiPP-encoding BGC from *G. apis*.

The computational analysis conducted in this study is sound and presents promising opportunities for future research in the fields of honey bee gut microbiota and natural product discovery. However, the experimental section is somewhat limited, and the conducted experiments do not strongly support the main conclusion regarding the protective roles of specific *G. apis* strains against *M. plutonius* infection in hive environments. Therefore, the title is biased, as the experimental findings do not show protection but only a reduction in *M. plutonius* abundance in the guts of adult bees. As the authors mention, adult bees only act as carriers of this bacterial pathogen, while larval bees are the ones affected. No experiments with larvae (or interactions between adult bees and larvae) are conducted to investigate whether this reduction in *M. plutonius* abundance in the adult bee gut would lead to a reduction in transmission to larvae and, consequently, protection. An in-depth discussion of this point is missing and should be extensively explored.

Other points that deserve an in-depth discussion include: What is the prevalence of *G. apis* encoding bioactive natural products in honey bees? What is the typical abundance of *M. plutonius* in asymptomatic adult bees and larvae? Would an approximate 10-fold reduction in *M. plutonius* abundance in the adult bee gut, as shown in Fig. 4d, be sufficient to prevent transmission from adults to larvae and thus protect the larvae from infection?

Moreover, it would be valuable to provide a more detailed explanation in the discussion regarding the rationale behind investigating how a Gram-negative bacterium, *G. apis*, could protect against infection by a Gram-positive bacterium, *M. plutonius*. As a reader, I became confused, especially considering the earlier discussion regarding the prevalence of lanthipeptides in Gram-positive bacteria like *Lactobacillus* spp. in the bee gut. Given the abundance of *Lactobacillus* as a member of the bee gut microbiota, why not investigate the natural products produced by this genus instead of *G. apis*? Additionally, *Lactobacillus* spp. are also associated with larvae, making them a great candidate for searching for new bioactive natural products.

Other comments:

Lines 3, 56, 109: Avoid abbreviating "honey bees" as "honey" after mentioning it alongside bumble bees. Simply use "honey bees" and "bumble bees"

Lines 29, 335: Ensure consistency in the use of the term "honey bee" or "honeybee" throughout the manuscript

Line 123: Correct to "*Bombilactobacillus mellifer*"

Line 40: Do not abbreviate "*Melisococcus plutonius*" here, as it is the first time it appears in the main text

Line 52: The reference to Raymann et al. 2017 does not link the immune system to antibiotic treatment and gut microbiota perturbation. Instead, Li et al. 2017 (10.1371/journal.pone.0187505) discusses this topic.

Lines 81-82: Reference 26 does not talk about a secondary metabolite involved in microbe-host interactions but rather microbe-microbe interactions.

Line 134-135: How many and what core protein sequences? This information should be included in Methods or Supplementary material

Line 256: The transition between the characterization of lanthipeptides from *Lactobacillus Firm5* and the focus on a different class of compounds in a different bacterial species is abrupt. Consider explaining better the rationale behind this shift in focus to *Gilliamella* in the experimental section.

Line 280 – You infected bees with 10^6 cells of *M. plutonius* but do not discuss the abundance of this pathogen usually detected in asymptomatic bees. What is the correlation here? This is important to show whether the impact observed in the in vivo experiment (a 10-fold reduction in CFUs) is field relevant.

For the in vivo experiments, have you checked mortality rates for bees exposed to *M. plutonius*? Have you considered doing these experiments with larvae?

Line 398 – You show that Lanthipeptide class III is primarily present in bumble bee associated *Lactobacillus*

Although you characterized the RiPP-like peptide produced by *G. apis* and performed some in vivo experiments with monocolonized bees, you did not confirm the in vivo production of this natural product, nor did you verify the monocolonization status of your bees.

Methods – None of the phylogenetic analysis are described in the Methods section.

Point-by-point response

We appreciated all the comments from the reviewers.

We have provided a point-by-point response to all suggestions.

All line numbers refer to those in the WORD file with track changes.

REVIEWER COMMENTS

Reviewer #1 (Remarks to the Author):

The paper by Lang et al. constitutes a well-executed piece of research on the biosynthetic diversity of honeybee gut microbiota, culminating in the identification of interesting RiPP natural products. The manuscript is well-written and accompanied by beautiful figures. Generally, I am very positive about the work, but I do have a number of feedback points that the authors could take along in their revisions:

Reply: We sincerely appreciate the reviewer's recognition of the significance of our work. In response to the reviewer's valuable feedback, we have incorporated more methodological information into the manuscript as suggested.

1. A description of methods for the whole-genome phylogeny is missing. Which model was used, which markers, which tree construction method?

Reply: Thank you for reviewer's feedback. We are sorry that we missed this part. Basically, all the strains analyzed in this study are derived from our previous publications (Su et al. 2021, *Microbiome*, PMID 34732245; Wu et al. 2021, *Microbiome*, PMID 34784973, Meng et al. 2022, *Microbiome*, PMID 36045431), and we also used the same methods for the whole-genome phylogeny. We have added a detailed description of the whole-genome phylogeny analysis in the Methods.

Line 480-492:

Phylogenomic analysis

We compiled a dataset comprising 449 genomes of isolates originating and 28 metagenome-assembled genomes from honey and bumble bee guts (Supplementary Data 1). To construct phylogenetic trees for all bacterial genomes from bee gut, we used PhyloPhlAn 3.0 employing 400 universal marker genes under the "diversity low" parameter⁶⁷, and the

maximum likelihood method was used. To create the whole-genome tree for the core bacterial genera, i.e., *Lactobacillus Firm5*, *Bombilactobacillus*, *Gilliamella*, *Snodgrassella*, and *Bifidobacterium*, we used Roary version 3.12.0⁶⁸ with the parameter '-blastp 75' to identify core single-copy genes shared among all strains. The resulting alignments of nucleotide sequences were concatenated, and the maximum-likelihood trees were inferred using FastTree version 2.1.10⁶⁹ with a generalized time-reversible (GTR) model. We utilized the iTOL web-based software⁷⁰ for the visualization of these phylogenetic trees.

2. For the metagenome data, a list of SRA accessions should be provided so that it is clear which samples were used and to which data columns in the SI tables they correspond.

Reply: Thank you for bringing up this important point. We have included the relevant SRA accessions in Supplementary Data 5.

3. Line 114-128: various statements are provided that remain vague ('mostly', 'mainly', etc.); please provide actual numbers wherever possible (also elsewhere in the manuscript).

Reply: Thank you for the reviewer's suggestion. We have replaced vague terms with specific numerical data throughout the manuscript (Line 116-122).

Line 116-122:

RiPPs were the most abundant BGC family (36.9%) in bee gut core bacterial members (Fig. 1a, b). They were mostly identified from the genomes of Gram-positives (83.6%), such as *Lactobacillus Firm5*, *Bombilactobacillus*, and *Bifidobacterium*. In *Gilliamella* strains, NRPS (34.9%) and RiPPs (20.7%) were the most abundant BGC families, while Terpene was mainly encoded by *Snodgrassella* (78.7%), *Commensalibacter* (11.2%), and *Bartonella* (8.7%) species (Fig. 1b, c).

4. Figure 1b: please use the same order of the genera in the color legend as the order in the tree

Reply: Thank you for reviewer's suggestion. We have modified Figure 1b to used the same order as the color legend.

5. Line 145-149: please indicate whether global or glocal mode was used

Reply: Thank you for reviewer's question. We used the 'global' mode for the BiG-SCAPE analysis and have indicated on Line 151.

Line 151: We used BiG-SCAPE³² to generate sequence similarity networks with default "global" mode for all BGCs annotated by antiSMASH from the bee gut bacterial genomes.

6. Line 153: how many BGCs had matches to MIBiG (again, please be quantitative) and which BGCs were they? Are they related to compounds that are expected to be found in this microbiome?

Reply: Thank you for pointing this out. There are 37 BGCs that are all from *Paenibacillus larvae*, which is a pathogenic bacterium affecting bee larvae, matched to 7 BGCs from MIBiG. They are all defined as 'T1PKS.NRPS'. We have added this information in the main text (Line 158-160).

Line 158-160: Only three defined BGCs (T1PKS.NRPS) from MIBiG fell into three GCFs with 37 BGCs from *P. larvae*, which is a pathogenic bacterium affecting bee larvae⁹.

7. For the novel characterized RiPP, it would be nice if the BGC were deposited into MIBiG and the structure elucidated (or potentially inferred from genome + MS/MS data, if possible). Is it predicted to be cyclic?

Reply: We appreciate the suggestions from the reviewer. The exact structure of the newly observed RiPP from *Gilliamella apis* was not characterized. We performed the genome mining algorithm with MetaMiner, a spectrometry-guided genome mining approach, to integrate the tandem mass spectra and genomic data to identify the RiPP-encoding BGC in *Gilliamella apis*. However, we totally agree with the reviewer that the elucidation of the RiPP structure is significant. We are now collaborating with chemists from other institutions and attempting to confirm the structures of the RiPPs,

also including the lanthipeptides from bee gut *Lactobacillus*, with more MS and NMR data. Indeed, we have identified five distinct planar structures of one NRPS from *Gilliamella* now. These results will be included in a separate study.

8. Figure 2c: the color scale has two gradients (red to white, white to blue), which I think is more logical for cases in which data can also be negative. I would recommend a single gradient.

Reply: We totally agree with the reviewer and have modified Figure 2c with a single gradient.

9. Figure 3: were there any signs of class V lanthipeptides (not detectable with antiSMASH, I think, but could be found using BLAST)?

Reply: Thank you for the suggestion. We did not detect Class V lanthipeptides from bee gut bacteria. We also followed the reviewer's suggestion to explore using BLAST. We used the cyclase (WP_171084029), LanK kinase (WP_171084025), and LanY lyase (WP_171084023) of the Class V lanthipeptide of *Streptomyces morookaensis* NRRL B-12429 as query sequences. We found no sequences from bee gut bacteria with a similarity > 20%.

10. Line 304-305: can the peptide-spectrum match be visualized perhaps, and added as an SI Figure?

Reply: Thank you for the suggestion. We have provided a raw dataset of the peptide-spectrum match from the MetaMiner analysis in Supplementary Table 1 now.

11. Line 462: why was antiSMASH 5.0 used? Historical reasons?

Reply: The reviewer is correct. We performed the antiSMASH analysis for the bee

gut database with 477 genomes using version 5.0 one years ago. We have performed the analysis with antiSMASH 7.0 again, while only 31 NRP-metallophore BGCs from *Snodgrassella* are predicted during this run. No more novel BGCs are detected with version 7.0 from *Gilliamella* or *Lactobacillus* Firm5, and the results remained consistent.

Spelling/grammar/style:

1. Line 20: 'the' -> 'a' (please also check usage of articles throughout)

Reply: Corrected (Line 20).

2. Line 80: the BGCs do not encode compounds, they encode their biosynthetic enzymes

Reply: Thank you for pointing it out, we have rewritten the sentence (Line 79-82).

Line 79-82: The secondary metabolites are often small chemical compounds produced by biosynthetic enzymes encoded by biosynthetic gene clusters (BGCs).

3. Line 86: 'The genome mining algorithm': which algorithm?

Reply: Thank you for pointing it out, we have rephrased the sentence. (Line 89).

4. Line 94: remove 'the'

Reply: Corrected (Line 96).

5. Line 100: 'the' -> 'a'

Reply: Corrected (Line 102).

Reviewer #2 (Remarks to the Author):

In this study, Lang et al. investigated the bee gut microbiota for small molecule biosynthetic gene clusters (BGCs) and identified numerous novel ones, including post-translationally modified peptides and lanthipeptides. They also observed significant variations in BGC distribution among bee species and geographic locations, reflecting strain-level differences within the bee gut microbiota. Additionally, they demonstrated that specific strains of *Gilliamella apis* possess a thiopeptide-like BGC, which exhibited antibacterial activity against the pathogenic bacterium *Melissococcus plutonius* in specific in vitro and in vivo experiments. They also employed a spectrometry-guided genome mining approach to discover the RiPP-encoding BGC from *G. apis*.

>**Reply:** We sincerely thank all the constrictive suggestions from the Reviewer and totally agree with all the raised concerns. Following the suggestion from Reviewer #2, we have modified the title and explained the experiment design of the pathogen concentration we used. We have also explained why we went into the natural products from the bee gut *Gilliamella* instead of *Lactobacillus* and have added the results of the incapability of inhibition by lanthipeptides from *Lactobacillus* in the Supplementary Figure. We have also added more citations and in-depth discussion regarding why the natural products from adult bee gut symbionts may help impede the pathogen transmission in the colony. We hope our responses will fulfill the requirements from the reviewer.

The computational analysis conducted in this study is sound and presents promising opportunities for future research in the fields of honey bee gut microbiota and natural product discovery. However, the experimental section is somewhat limited, and the conducted experiments do not strongly support the main conclusion regarding the protective roles of specific *G. apis* strains against *M. plutonius* infection in hive environments. Therefore, the title is biased, as the experimental findings do not show protection but only a reduction in *M. plutonius* abundance in the guts of adult bees. As the authors mention, adult bees only act as carriers of this bacterial pathogen, while larval bees are the ones affected. No experiments with larvae (or interactions between adult bees and larvae) are conducted to investigate whether this reduction in *M. plutonius* abundance in the adult bee gut would lead to a reduction in transmission to larvae and, consequently, protection. An in-depth discussion of this point is missing and should be extensively explored.

>**Reply:** We sincerely appreciated the constructive suggestions from the reviewer. The reviewer is correct that *M. plutonius* mainly affects the larvae but not the adult bees. In this study, we found that the natural products from the adult bee gut bacteria, *Gilliamella*, potentially inhibit the proliferation of *M. plutonius* in vivo and in vitro. We totally agree with the reviewer that it would be perfect to perform the experiments with bee larvae. But it is difficult to set up a “gnotobiotic colony” with adult bees and/or larvae mono-colonized with specific bacterial strains, as we aimed

to observe the natural products from specific bacterial strains.

EFB is purely a disease of the digestive tract of the larvae (Tarr 1938, doi: 10.1111/j.1744-7348.1938.tb02356.x). Previous studies show that the first step in EFB infection is an asymptomatic colonization of bee larvae due to the food transmission by adult nurse bees (Bailey 1959, Journal of insect pathology. 1 (1), pp. 80-85; Bailey 1960, Journal of Insect Pathology. 1960;2(2):67–83.). Additionally, Bailey (Bailey 1960, Journal of Insect Pathology. 1960;2(2):67–83.1960; Bailey 1983, doi: 10.1111/j.1365-2672.1983.tb02648.x) suggested that adult bees always remove infected larvae to achieve an adequate food supply supporting the survival of uninfected larvae. McKee et al. (2003, doi:10.1051/apido:2002047) demonstrated that the adult worker bees removing infected larvae may act as vectors of *M. plutonius* cells via food transmission to healthy broods within and between colonies. Thus, reduction in *M. plutonius* abundance in the adult bee gut may help with a reduction in transmission to larvae.

We totally agree with the reviewer that our title is biased and have modified it to “Natural products from gut symbionts as antimicrobial agents against *Melissococcus plutonius* in honeybees”. We have also added more discussion in the text (Line 456-478).

Line 456-478

EFB is purely a disease of the digestive tract of the larvae⁶², while the first step in EFB infection is an asymptomatic colonization of larvae due to the food transmission by adult nurse bees^{63,64}. Additionally, adult workers removing infected larvae may transfer pathogens to healthy broods within colonies⁶⁵ and even between colonies through robbing and drifting⁶⁶. Erban et al.¹² showed that *M. plutonius* reached $\sim 10^6$ CFU/bee in the adult bees from colonies with clinical symptoms, which is 75-fold-higher than those from asymptomatic colonies. Our results showed that gnotobiotic bees mono-colonized with *G. apis* stain B14384H2 can achieve ~ 100 -fold reduction of *M. plutonius* (on average, from 2.12×10^8 to 6.22×10^6 , $p = 0.0079$). However, future studies on the interactions between adult bees and larvae are needed to investigate the role of potential bioactive molecules in the infection control. Honeybee gut bacteria adapting to the gut environment and competing against non-native microorganisms may drive the selection of BGCs that produce highly efficacious

natural products. Our study highlights the distribution of BGCs with uncharacterized functions in the bee gut microbiome and provides novel insights into the potential novel therapeutic agents for bee diseases.

Other points that deserve an in-depth discussion include: What is the prevalence of *G. apis* encoding bioactive natural products in honey bees? What is the typical abundance of *M. plutonius* in asymptomatic adult bees and larvae? Would an approximate 10-fold reduction in *M. plutonius* abundance in the adult bee gut, as shown in Fig. 4d, be sufficient to prevent transmission from adults to larvae and thus protect the larvae from infection?

>Reply: We thank the reviewer for pointing these out. There are basically three species of *Gilliamella* in the honeybee gut, *Gilliamella apis*, *Gilliamella apicola*, and one undefined lineage (Li et al. 2022, doi:10.1073/pnas.2115013119). We found that only strains from the *Gilliamella apis* s encode the bioactive natural products, but the other two species do not. Typically, *Gilliamella apis* and *Gilliamella apicola* are the two predominant species in the bee gut (Wu et al. 2021, PMID 34784973; Ellegaard et al. 2020, doi: 10.1016/j.cub.2020.04.070). We have modified the text to better illustrate this point (Line 433-436).

Erban et al. (2017, doi:10.7717/peerj.3816) showed that the adult bees from colonies with clinical symptoms of EFB exhibited a 75-fold-higher relative abundance of *M. plutonius* than those from asymptomatic colonies. Based on the 16S rRNA sequencing result from Erban et al. (2017), the relative abundance of *M. plutonius* is ~3%. Considering the absolute number of bacteria in the adult bees ($\sim 10^8$ CFU/bee), the load of *M. plutonius* would be around 10^6 CFU/adult bee. Therefore, we set up our experiment by feeding individual bees with 10^6 CFU manually. However, potentially due to that we used microbiota-free bees in this experiment, *M. plutonius* grew to $\sim 10^8$ CFU/bee.

In an epidemiological study, Roetschi et al. (2008, doi:10.1051/apido:200819) developed a real-time PCR assay for *M. plutonius*. They found that the load of *M. plutonius* is $\sim 5 \times 10^4$ CFU in workers from brood nests of colonies without clinical symptoms and have suggested this quantification threshold for a diagnostic tool to screen colony health status. Our results showed that gnotobiotic bees mono-colonized with *G. apis* strain W8126 resulted in ~ 10 -fold reduction in *M. plutonius* abundance, while *G. apis* stain B14384H2 can achieve a 100-fold reduction (on

average, from 2.12×10^8 to 6.22×10^6 , $p=0.0079$).

However, we agree with the reviewer that it is not sure whether this reduction is sufficient to prevent the transmission and subsequently protect the larvae from infection. We have added more discussion in the text (Line 456-478).

Line 433-436:

Here, we investigated that strains from *G. apis*, but not *G. apicola*, inhibit the growth of *M. plutonius*, while they are both predominant *Gilliamella* species in the honeybee gut^{17,36}. *G. apis* strains harbor the BGCs of potential thiopeptide, which react as protein synthesis inhibitors by binding to the bacterial ribosome.

Moreover, it would be valuable to provide a more detailed explanation in the discussion regarding the rationale behind investigating how a Gram-negative bacterium, *G. apis*, could protect against infection by a Gram-positive bacterium, *M. plutonius*. As a reader, I became confused, especially considering the earlier discussion regarding the prevalence of lanthipeptides in Gram-positive bacteria like *Lactobacillus* spp. in the bee gut. Given the abundance of *Lactobacillus* as a member of the bee gut microbiota, why not investigate the natural products produced by this genus instead of *G. apis*? Additionally, *Lactobacillus* spp. are also associated with larvae, making them a great candidate for searching for new bioactive natural products.

>Reply: We thank the reviewer for pointing this out. Actually, we also tested the potential of the culture supernatant of *Lactobacillus* Firm5 as we did with *Gilliamella* spp., but it did not show obvious inhibition. Now, we have also added these results in the text (Line 281-283) and in the Supplementary Figure 9.

Previous studies with gut microbiome identified many natural products from Gram-negatives, and these natural products can also inhibit the growth of some Gram-positives. For example, Massetolide E produced by *Pseudomonas fluorescens* can inhibit the growth of *Bacillus thuringiensis* (doi: 10.1016/j.cub.2019.01.050). We have modified the discussion on Line 430-455.

We totally agree with the reviewer that it would be valuable to investigate the natural products produced by *Lactobacillus* considering its abundance in the bee gut and its association with the larvae. However, the natural products produced by the bee gut *Lactobacillus* belong to the lanthipeptides, which are ribosomally synthesized

peptides with posttranslational modification, including dehydration and cyclization, to form thioether cross-links. This class of compounds is always low-yielding, making the purification and identification challenging. Thus, the BGCs encoding lanthipeptide are always cloned into vectors, and the products can be obtained from engineered *E. coli* strains. Indeed, we are attempting to obtain the lanthipeptides from bee gut *Lactobacillus* by cloning the BGCs into *E. coli*. We are trying to first illustrate the posttranslational modification process of these lanthipeptides, as they are uncommon Class III lanthipeptides different from any defined lanthipeptide so far. These results will be included in a separate study.

Line 281-283:

In addition, we also tested the activity of the supernatants of *Lactobacillus* Firm5 encoding lanthipeptide BGCs, but no inhibition was observed (Supplementary Fig. 9).

Line 430-455:

Additionally, RiPPs are a promising source of alternative antimicrobials, especially as antibiotic resistance has become a growing crisis⁵⁶. Specifically, thiopeptides are a large group of structurally complex natural products that always show strong antibacterial activity against Gram-positives⁵⁷. Here, we investigated that strains from *G. apis*, but not *G. apicola*, inhibit the growth of *M. plutonius*, while they are both predominant *Gilliamella* species in the honeybee gut^{17,36}. *G. apis* strains harbor the BGCs of potential thiopeptide, which react as protein synthesis inhibitors by binding to the bacterial ribosome. Since the binding sites of thiopeptide are distinct from used antibiotics, they may overcome the existing antibiotic resistance in pathogens⁵⁸. For example, thiazomycin show potent activity against the multidrug-resistant *Mycobacterium tuberculosis*⁵⁹, and nosiheptide effectively inhibits the methicillin-resistant *Staphylococcus aureus* and vancomycin-resistant *Enterococcus*⁶⁰. The antiSMASH analysis identified the BGC of *G. apis* containing a set of genes, including the core biosynthetic genes for thiopeptide-like BGC, such as the azoline-forming YcaO cyclodehydrating Cys, Ser, and Thr residues, radical SAM-dependent methyltransferase, ABC transporters, and the peptidases⁶¹. However, this BGC from *Gilliamella* differs significantly from the currently known RiPPs in the database. Future investigations on the post-translational modification and maturation process are needed to solve the structure of the novel RiPP from bee gut symbionts.

Other comments:

1. Lines 3, 56, 109: Avoid abbreviating "honey bees" as "honey" after mentioning it alongside bumble bees. Simply use "honey bees" and "bumble bees"

Reply: Corrected.

2. Lines 29, 335: Ensure consistency in the use of the term "honey bee" or "honeybee" throughout the manuscript

Reply: Corrected.

3. Line 123: Correct to "Bombilactobacillus mellifer"

Reply: Corrected (Line 126).

4. Line 40: Do not abbreviate "Melisococcus plutonius" here, as it is the first time it appears in the main text

Reply: Corrected (Line 41).

5. Line 52: The reference to Raymann et al. 2017 does not link the immune system to antibiotic treatment and gut microbiota perturbation. Instead, Li et al. 2017 (10.1371/journal.pone.0187505) discusses this topic.

Reply: We have replaced it with the designated reference.

6. Lines 81-82: Reference 26 does not talk about a secondary metabolite involved in microbe-host interactions but rather microbe-microbe interactions.

Reply: This sentence has been rephrased (Line 84).

7. Line 134-135: How many and what core protein sequences? This information should be included in Methods or Supplementary material

Reply: Thank you for pointing this out. We have added more detailed information in the figure legend (Figure 1) and the Methods (Line 480-492) now.

Line 480-492:

Phylogenomic analysis

We compiled a dataset comprising 449 genomes of isolates originating and 28 metagenome-assembled genomes from honey and bumble bee guts (Supplementary Data 1). To construct

phylogenetic trees for all bacterial genomes from bee gut, we used PhyloPhlAn 3.0 employing 400 universal marker genes under the "diversity low" parameter⁶⁷, and the maximum likelihood method was used. To create the whole-genome tree for the core bacterial genera, i.e., *Lactobacillus Firm5*, *Bombilactobacillus*, *Gilliamella*, *Snodgrassella*, and *Bifidobacterium*, we used Roary version 3.12.0⁶⁸ with the parameter '-blastp 75' to identify core single-copy genes shared among all strains. The resulting alignments of nucleotide sequences were concatenated, and the maximum-likelihood trees were inferred using FastTree version 2.1.10⁶⁹ with a generalized time-reversible (GTR) model. We utilized the iTOL web-based software⁷⁰ for the visualization of these phylogenetic trees.

8. Line 256: The transition between the characterization of lanthipeptides from *Lactobacillus Firm5* and the focus on a different class of compounds in a different bacterial species is abrupt. Consider explaining better the rationale behind this shift in focus to *Gilliamella* in the experimental section.

Reply: We thank the constructive suggestion from the reviewer. We also tried with the supernatant of *Lactobacillus* culture, which did not show obvious inhibition. We have added these results in the text (Line 281-283) and in the Supplementary Figure 9.

9. Line 280 – You infected bees with 10^6 cells of *M. plutonius* but do not discuss the abundance of this pathogen usually detected in asymptomatic bees. What is the correlation here? This is important to show whether the impact observed in the in vivo experiment (a 10-fold reduction in CFUs) is field relevant.

Reply: We thank the reviewer for pointing this out. We have added more discussion in the text now (Line 456-478).

Erban et al. (2017, doi:10.7717/peerj.3816) showed that the adult bees from colonies with clinical symptoms of EFB exhibited a 75-fold-higher relative abundance of *M. plutonius* than those from asymptomatic colonies. Based on the 16S rRNA sequencing result from Erban et al. (2017), the relative abundance of *M. plutonius* is ~3%. Considering the absolute number of bacteria in the adult bees ($\sim 10^8$ CFU/bee), the load of *M. plutonius* would be around 10^6 CFU/adult bee. Therefore, we set up our experiment by feeding individual bees with 10^6 CFU manually. However, potentially due to that we used microbiota-free bees in this experiment, *M. plutonius* grew to $\sim 10^8$ CFU/bee.

In an epidemiological study, Roetschi et al. (2008, doi:10.1051/apido:200819)

developed a real-time PCR assay for *M. plutonius*. They found that the load of *M. plutonius* is $\sim 5 \times 10^4$ CFU in workers from brood nests of colonies without clinical symptoms and have suggested this quantification threshold for a diagnostic tool to screen colony health status. Our results showed that gnotobiotic bees mono-colonized with *G. apis* strain W8126 resulted in ~ 10 -fold reduction in *M. plutonius* abundance, while *G. apis* strain B14384H2 can achieve a 100-fold reduction.

10. For the in vivo experiments, have you checked mortality rates for bees exposed to *M. plutonius*? Have you considered doing these experiments with larvae?

Reply: We thank the reviewer for pointing this out. We followed the mortality rates but no bees died during the infection, but there is no effect. Previous studies suggest that *M. plutonius* do not affect adult bees (doi: 10.1016/j.jip.2009.06.016; doi: 10.3390/insects12020150; Andrea Cecchini P.H., Dietemann V., Charrière J.-D., Grossar D. The Influence of European Foulbrood on the Mortality of Adult Honeybees; Proceedings of the 7th European Conference of Apidology; Cluj-Napoca, Romania. 7–9 September 2016.). We agree with the reviewer that it would be perfect to do the experiments with larvae, but it is difficult to build a “gnotobiotic colony” with bees mono-colonized with specific bacteria strains. It is also due to that *Gilliamella* is not a typical bacterium in the bee larvae. We are planning to perform the experiments with larvae when we obtain the natural products from the *Lactobacillus* strains. We sincerely appreciated the suggestions from the reviewer.

11. Line 398 – You show that Lanthipeptide class III is primarily present in bumble bee associated *Lactobacillus*

Reply: We thank the reviewer for pointing this out. We have rewritten this sentence (Line 404-405).

12. Although you characterized the RiPP-like peptide produced by *G. apis* and performed some in vivo experiments with monocolonized bees, you did not confirm the in vivo production of this natural product, nor did you verify the monocolonization status of your bees.

Reply: We thank the reviewer for pointing this out. We checked the mono-colonization status of the treat bees, and this result was added in the Supplementary Figure S10.

We totally agree with the reviewer that it is significant to confirm the in vivo

production of the identified natural product from *Gilliamella*. However, there are many different classes of natural products with broad chemical diversity. In addition, the concentration of the bacterial natural products is low under in vivo conditions. To discriminate the novel thiopeptide from *Gilliamella*, we must first enrich the products and illustrate the exact chemical structure using Nuclear Magnetic Resonance (NMR) and Mass Spectrometry (MS). Then, we can track the concentration of the designated compounds in the bee gut. We are not collaborating with chemists from other institutions to systematically purify and characterize the structures of various natural products from the *Gilliamella* and *Lactobacillus* strains, these results will be involved in a separate study.

13. Methods – None of the phylogenetic analysis are described in the Methods section.

Reply: Thank you for pointing this out. This is also mentioned by Reviewer #1. We have added detailed information of the phylogenetic analysis in the Methods now (Line 480-492).

REVIEWERS' COMMENTS

Reviewer #1 (Remarks to the Author):

I am generally happy with the changes made during the revisions based on my comments.

Only two minor points remain from my side:

- On first sight, it seems that in Figure 1B, the taxonomic clade at 7 o'clock in the tree (in red) is not represented in the color legend, unless this is the same 'Others' color again (which on second thought seems to be true). Still, I was slightly confused.
- For the 7 MIBiG matches, it would be very helpful for the readers to know which compounds (BGC products) they are associated with.

Reviewer #2 (Remarks to the Author):

The authors have addressed my previous comments, significantly improving the manuscript's overall quality. They have also acknowledged the need for additional work to validate their main findings and statements regarding the protective effects of *G. apis* against *M. plutonius*, but it appears that these experiments are either ongoing or planned for separate publication. Therefore, I have no further comments at the moment.

Point-by-point response

REVIEWER COMMENTS

Reviewer #1 (Remarks to the Author):

I am generally happy with the changes made during the revisions based on my comments. Only two minor points remain from my side:

- On first sight, it seems that in Figure 1B, the taxonomic clade at 7 o'clock in the tree (in red) is not represented in the color legend, unless this is the same 'Others' color again (which on second thought seems to be true). Still, I was slightly confused.

Reply: We thank the reviewer for the constructive suggestions. We have modified the colors of “*Frischella*” and “Others” to avoid any confusion.

- For the 7 MIBiG matches, it would be very helpful for the readers to know which compounds (BGC products) they are associated with.

Reply: We thank the reviewer for the constructive suggestions. We have rewritten this sentence:

Line 145: Only two defined BGCs, paenilamicin (BGC0001033) and sevadicin (BGC0000426) from MIBiG fell into three GCFs with 37 BGCs from *P. larvae*, which is a pathogenic bacterium affecting bee larvae⁹.

Reviewer #2 (Remarks to the Author):

- The authors have addressed my previous comments, significantly improving the manuscript's overall quality. They have also acknowledged the need for additional work to validate their main findings and statements regarding the protective effects of *G. apis* against *M. plutonius*, but it appears that these experiments are either ongoing or planned for separate publication. Therefore, I have no further comments at the moment.

Reply: We thank the reviewer for providing valuable comments.